# Classifying the Unclassified: A Phage Classification Method

**DOI:** 10.3390/v11020195

**Published:** 2019-02-24

**Authors:** Cynthia Maria Chibani, Anton Farr, Sandra Klama, Sascha Dietrich, Heiko Liesegang

**Affiliations:** Institute for Microbiology and Genetics, Georg-August University Goettingen, Grisebachstr. 8, 37077 Goettingen, Germany; cchiban@gwdg.de (C.M.C.); anton.farr@stud.uni-goettingen.de (A.F.); sandra.klama@gwdg.de (S.K.); sascha.dietrich@uni-wuerzburg.de (S.D.)

**Keywords:** Hidden Markov Models, *Vibrionaceae*, vibriophages, *Inoviridae*, *Myoviridae*, *Podoviridae*, *Siphoviridae*, phages, classification, protein coding sequences

## Abstract

This work reports the method ClassiPhage to classify phage genomes using sequence derived taxonomic features. ClassiPhage uses a set of phage specific Hidden Markov Models (HMMs) generated from clusters of related proteins. The method was validated on all publicly available genomes of phages that are known to infect *Vibrionaceae*. The phages belong to the well-described phage families of *Myoviridae*, *Podoviridae*, *Siphoviridae*, and *Inoviridae*. The achieved classification is consistent with the assignments of the International Committee on Taxonomy of Viruses (ICTV), all tested phages were assigned to the corresponding group of the ICTV-database. In addition, 44 out of 58 genomes of *Vibrio* phages not yet classified could be assigned to a phage family. The remaining 14 genomes may represent phages of new families or subfamilies. Comparative genomics indicates that the ability of the approach to identify and classify phages is correlated to the conserved genomic organization. ClassiPhage classifies phages exclusively based on genome sequence data and can be applied on distinct phage genomes as well as on prophage regions within host genomes. Possible applications include (a) classifying phages from assembled metagenomes; and (b) the identification and classification of integrated prophages and the splitting of phage families into subfamilies.

## 1. Introduction

Phages, defined as viruses that infect bacteria, are the most abundant biological entities known so far [1,2]. The taxonomic classification of viruses and naming of virus taxa is maintained by the International Committee on Taxonomy of Viruses (ICTV) [3] and the Bacterial and Archaeal Subcommittee (BAVS) within the ICTV that focuses on phages. The system is based on the evaluation of a variety of phage properties including the molecular composition of the virus genome (ss/ds, DNA, or RNA), the structure of the virus capsid and whether or not it is enveloped, the host range, pathogenicity, and sequence similarity [4,5]. Based upon these different properties the ICTV established a highly valuable and widely accepted Virus taxonomy. Considering the complexity of features that contribute to the taxonomy of a phage a comprehensive guideline has been published by Adriaenssens and Brister [6]. However, due to the availability of Next Generation Sequencing (NGS)-technologies an increasing amount of genomic and metagenomic sequence data is available that include complete as well as fragments of so far unknown phage genomes [7,8]. Unfortunately, a systematic classification of these genomes into the ICTV scheme is impossible due to the lack of corresponding biological and experimental data [4,9,10]. So for that matter, a taxonomic characterization based on the phages genome sequence information has become indispensable [5].

Many attempts at creating viral phylogenetic trees have failed due to the lack of a universal marker gene and the high mosaic structure of phages [11]. Sequence-based phylogenetic analysis procedures like 16S and multi locus sequence typing (MLST) [12,13,14] are based on the existence of an orthologous marker molecule shared among monophyletic entities. The finding that phages are a polyphyletic group of biological entities results in the finding that orthologous markers are available only within the monophyletic subgroups of phages [15]. Consequently, sequence alignment and similarities based approaches using single selected marker molecules have been designed for phage classifications restricted to closely related phage taxa [16]. Clustering techniques for viral classification have been applied by several authors and confirmed that comparative sequence analysis is effective [11,17,18,19]. Deschavanne et al. [20] demonstrated that genomic signatures based on oligomer composition are effective to determine the phylogenetic distance of closely-related phages and their hosts, as well as within the phages preying on related hosts. The investigation revealed that in the case of temperate phages, the amelioration process [21] interferes with the calculation of phylogenetic distances between phages. Rohwer and Edwards used the presence and the similarity of shared proteins to generate a phage proteomic tree using 105 complete sequenced genomes [22]. This approach is robust towards dynamic changes in the nucleotide composition. However, proteomic trees are limited in cases where the BLASTP based similarity determination is challenged by distantly related protein sequences. Bolduc et al. [23] introduced vConTACT, a tool that uses protein clusters and bipartite network-based distances to assign a given dsDNA phage genome to a taxon.Aiewsakun et al. [24] demonstrated that the Genome Relationship Applied to Virus Taxonomy (GRAViTy) software platform, which is designed for eukaryotic virus genomes, performs well on monophyletic subfamilies of viruses that infect bacteria and archaea. GRAViTy uses composite generalized Jaccard (CGJ) distances based on shared genomic features to determine the genetic relatedness of a given set of virus genomes.

The development of bioinformatics methods to recognize and characterize genomics elements is strongly supported if a well-described sample dataset is available. In the case of our project, we selected vibriophages, i.e., phages that infect *Vibrionaceae*, as a training dataset. Vibriophages are known as an important driving force of the evolution of *Vibrionaceae*, contributing to the emergence of virulence and the ecological success of this genus [25]. In the case of *Vibrio cholera*, the causative of the pandemic disease cholera (WHO newsletter 2018), the virulence of the bacterium is encoded by viral genes of the phage. Due to its medical importance, it is a well investigated example of how phages contribute to the evolution and the virulence of bacterial hosts [26,27,28]. *Vibrionaceae* include in addition a number of important fish pathogens, where integrated prophages have been shown to contribute to the virulence of the strains, and thus leading to great economic losses [29]. *Inoviridae*, which comprises the CTX-phage of *V. cholera* [30], as well as the filamentous M13 phage [31], are among the best-investigated phages that have been studied for more than VI decades [28]. Due to the medical and economic importance and the in detail molecular biological knowledge on *Inoviridae*, a substantial amount of sequencing data on *Vibrionaceae* and Vibriophages is available. Castillo et al. [21] have recently estimated that there exist 5674 prophage-like elements within 1874 published *Vibrio* genome sequences, and that 45% of the strains harbor prophages of the family *Inoviridae*, that contribute by lysogenic conversion, with the Zonaoccludens toxin (Zot), to the virulence of *Vibrionaceae*. Multiple studies have shown the presence of *Caudovirales* in addition to *Inoviridae* phages in *Vibrio* species [32,33,34]. This makes this group an excellent test case for a sequence based characterization method and a potential identification of phages.

Hidden Markov Model (HMM) based search and clustering methods proved to be efficient for the characterization of protein families, as well for the taxonomic characterization of corresponding genes [15,35]. Here, we present a case study that investigates profiles of combined HMMs derived from related dsDNA and ssDNAphage genomes, and their efficiency characterize and potentially identify members of four well-described families of vibriophages. We demonstrate that a method based exclusively on genome sequences achieves a classification of phages that is consistent with the ICTV standards. Furthermore, a genomic analysis of the profile HMM characterized genomes, reveals details and relation of phages corresponding to their phylogenetic distance and their host range.

## 2. Materials and Methods

### 2.1. Data Sources

#### 2.1.1. Phages

Various phage datasets have been used in this study. Firstly, publicly available Vibriophages sequence datasets were downloaded from NCBI nucleotide database by keyword search; date of accession 13 February 2018. In total numbers, 159 phage genomes out of which 58 were unclassified, 19 *Inoviridae*, 37 *Myoviridae*, 42 *Podoviridae*, and 15 *Siphoviridae* genomes infecting *Vibrio* genomes were downloaded (Appendix A). This dataset was split into a classified for the generation of HMM models and an unclassified dataset to which the HMMs were applied for classification purposes.

Secondly, a set of 19 experimentally proven and sequenced *Inoviridae* phages derived from a genome sequencing project on 9 *V. alginolyticus* strains and 1 *V.typhli* strain isolated from Pipefish [29], was used for validation of the *Inoviridae* generated HMMs.

Lastly, in order to test the limitations of the method, sequence datasets of the four phage families were downloaded from the Millard lab database (http://millardlab.org/ bioinformatics /bacteriophage-genomes/); date of accession March 2018. In total numbers, 119 *Ino*-, 1766 *Myo*-,1066 *Podo*- *and* 3466 *Siphoviridae* were downloaded (Appendix A).

#### 2.1.2. Host Genomes

In order to test the generated HMMs for phage identification, 154 closed *Vibrio* genomes publicly available (Appendix A) were downloaded from NCBI by keyword search; date of accession 18.06.2018. In total numbers, 154 *Vibrio* genomes out of which 39 were *V. cholerae*, 22 *V. parahaemolyticus*, 15 *V. vulnificus*, 13 *V. alginolyticus*, 13 *V. anguillarum*, 9 *V. campbellii*, 5 *V. natriegens*, 4 *V. harveyi*, 4 *V. coralliilyticus*, and other *Vibrio* species were downloaded.

### 2.2. Data Preparation

For each genbank file, a multi-FASTA file containing all annotated coding sequences was created. The collected protein sequences were concatenated and clustered with the Markov clustering algorithm (MCL) [36]. CD-hit [37] (V4.5.4) was used to remove redundant proteins. In addition, information on classification, host, phage size, isolation source was extracted from each genbank file.

### 2.3. Profile HMM Construction

Produced multi-sequence alignment files were used to build profile HMMs [38], using the “hmmbuild” command available as part of the HMMER (v3.1b1) package. Subsequently, sensitive profile HMMs were created out of a minimum of five clustered proteins. Removed proteins were stored for later refinement steps. The command “hmmpress” was used to create binary compressed data files (.h3m, .h3i, .h3f, and .h3p) from a profile HMM. These binary files were used to look for orthologous protein hits in the scanned dataset. The scanned input dataset was used to map hit to the phage family proteins they were derived from. The function “hmmemit” was used to create a consensus sequence from a generated profile HMM. This consensus sequence is closest in similarity to the majority of sequences used to create the respective HMM.

### 2.4. Profile HMM Refinement

Using “BLASTP” to align each protein of a cluster against the consensus sequence, and by specifying the output table to feature the coverage of each sequence compared to the consensus, the coverage was compared with the user-specified threshold(standard <50%). Proteins not reaching the threshold were removed. Created profile HMMs were used to scan the original master-FASTA. Proteins were refined according to hits of (a) proteins removed due to redundancies, (b) proteins used to create the HMMs themselves, and (c) not yet assigned proteins. Proteins which are hit and have not yet been assigned were added to the profile HMM. Proteins that were used to create the HMM and were not hit were removed from the profile HMM. Proteins that are hit but were removed previously due to redundancies were not added. Whenever multiple HMMs hit the same sets of proteins as well as their inputs, they were merged. Otherwise, HMMs were not merged. Refined HMMs were used to rescan the input master-FASTA and if needed refinement steps of merging were repeated until no changes occured.

### 2.5. CDS Prediction and Additional HMM Refinement

Nucleotide sequences between predicted coding sequences (CDS) were extracted from each genbank file and were translated into an amino acid sequence. Generated refined HMMs were used to scan the translated regions. A 50% alignment coverage, a negative bit-score value and an E-value over 1.5 × 10^−8^ were used as cut-offs to filter the generated hmmscan output. Hits passing the filtered cut-offs were integrated in the multiple sequence alignment (MSA) input per HMM and HMMs were rebuilt with the updated MSA. The regenerated HMMs were used to rescan the input phage master-FASTA files in order to compare HMMs performance when generated based on the original genbank files and the HMMs generated based on improved genomes. The generated HMMs can be downloaded at http://appmibio.uni-goettingen.de/index.php?sec=sw.

### 2.6. Software Tools

PHASTER was used to scan all 154 *Vibrio* gbk files (Appendix A) for the identification of integrated phages. Visualization was performed using R version 3.2.3 in Rstudio version 1.1.383 and using the R package “ggplot2” version 3.0.0 unless stated otherwise.

## 3. Results and Discussion

### 3.1. Phage Protein Families and Profile HMMs

To generate the initial set of HMMs, the protein sequences of all 110 available genomes known to infect *Vibrionaceae* were extracted. The data consists of the proteins from 19 *Ino-*, 35 *Myo-*, 42 *Podo-*, and 14 *Siphoviridae* phages. To ensure the internal model diversity, redundant sequences were removed and the remaining protein sequences were clustered with the *Markov cluster algorithm* (MCL) [37]. Models generated from clusters of five or more diverse sequences per protein family were evaluated for their taxonomic specificity (Table 1). In cases where models generated significant better hits against proteins of the phage taxon from which they have been encoded, the HMMs were considered as taxonomic indicators of the phage family.

The procedure resulted in 401 HMMs representing taxonomic indicative profile HMMs. In total 9 HMMs specific for *Ino-*, 242 for *Myo-*, 96 for *Podo-*, and 54 for *Siphoviridae* were identified as taxonomic indicators. The proteins used to generate refined HMMs per phage family are summarized in Appendix A. Note that, due to the lack of a sufficient number of diverse protein sequences, for some protein families no profile HMMs has been generated. 

### 3.2. Taxon Specificity of the Protein Family Models

To evaluate the discriminative power of a protein family based taxonomy, the profile HMMs were applied on three different data sets. (I) HMM scan to classify genomes of bacteriophages, known to prey on *Vibrionaceae*, into taxonomic groups consistent to the rules defined by the ICTV. (II) A scan of all proteins encoded by host genomes to investigate the, performance of the method to classify as well as potentially identify integrated prophages. In this test, host genomes with known biologically active vibriophages were used as proof of principle. (III) Scan of proteins of all known phage genomes from the taxa *Ino*-, *Myo*-, *Podo*-, and *Siphoviridae*.

### 3.3. Consistency of Taxon-Specific HMMs

The refined profile HMMs, derived out of the four phages families, were used in scans against all 4630 proteins encoded by the 110 phage genomes (Figure 1).

An application of the HMM profiles on the input phage proteome sequences revealed that the vast majority of the proteins (83.45%) match exclusively the taxon specific HMMs from the corresponding phage family. However, there was a number of 16.37% cross matches between the different families within the *Caudovirales* models, which indicates that the investigated phage genomes might represent a monophyletic group within the *Caudovirales* [39]. In contrast, 0.17% cross-matches occurred between *Caudovirales* and *Inoviridae* and thus support the hypothesis that there is gene exchange between these not monophyletic taxa [39].

#### 3.3.1. Inoviridae

In the case of the *Inoviridae* HMMs scanning *Caudovirales* proteomes, all HMMs match exclusively proteins encoded in *Inoviridae* genomes, except of one case which has an e-value of 2.9 × 10^−6^ to a protein annotated as “putative streptomycin biosynthesis operon regulatory protein (YP_009021749.1)”. While *Caudovirales* HMMs scanning *Inoviridae* proteomes, all HMMs match exclusively proteins encoded in *Caudovirales* except in seven cases where the e-value ranged between 1.5 × 10^−8^ and 7.8 × 10^−5^ to proteins annotated as ”hypothetical protein” and “RstR” (Appendix A). The low number of cross matches between *Inoviridae* and *Caudoviridae* is due to the phenotypical unique features of filamentous phages in contrast to tailed phages [40,41]. However, cross match hits may as well reflect genes that have been exchanged between *Inoviridae* and *Caudovirales* by a horizontal gene transfer (HGT) event [11]. Under this condition, the lower quality of the match score would reflect the time that the proteins evolved after the HGT-event within their separate viral host genomes.

#### 3.3.2. Caudovirales

In case of *Caudovirales,* scans of HMMs against their encoded proteins lead to a considerable number of cross matches (16.37%, 758 out of 4630). The proteins are related to basic phage functionality that are expected to be encoded by genomes of tailed phage like DNA polymerase, DNA replication initiation protein, ribonucleases, helicases, endonucleases, ligases, terminase, and phage tail proteins, as well as hypothetical proteins (Appendix A). However, the taxon derived models scored better against taxon encoded proteins. The type of the proteins and the correlation of HMM scores indicate that the matches are due to the shared genes with a common phylogenetic history of the tailed phages [11] and not to false positive recognition event of the HMMs.

To further explore vibriophages of the three *Caudovirales* groups, genome alignments were performed revealing that the virus genomes have a host specific diversity (Figure 2).

Genome alignments of the three *Caudovirales* in most of the cases revealed extended sequence similarities according to the BLAST algorithm within members of the taxonomic groups. Note that the BLAST algorithm is considerably less sensitive to identify distant similar sequences in comparison to the profile HMM [15]. This reduced sensitivity is the reason why BLAST based algorithms miss the taxonomic proximity of the *Myoviridae* phages 54-7, I1895-B1, and helene 12B3 as well as between the *Myoviridae* Eugene 12A10, RYC, and ICP1_2004_A (Figure 2). However, in most of the cases of *Myo*- and *Siphoviridae*, all members exhibit different degrees of similarities over the complete genome sequences and thus support the statement that the families are monophyletic [24,42]. However, within the *Podoviridae*, the comparison revealed four subgroups that did not show pronounced sequence.

### 3.4. Classification of Unclassified Phages

To examine the generated profile HMMs with regard to their application as a means of genome sequence based classification of bacteriophages, HMMs derived out of the four different bacteriophage families were used on to scan the proteomes of 58 published but taxonomically unclassified *Vibrio*-phages (Figure 3, Appendix A). The details of the HMM scan are summarized in Appendix A.

#### Taxonomic Assignments of Tailed Phages

For some of the investigated phage genomes, a taxonomic assignment based on experimental data is available. Zahid et al. [43] classified the vibriophages JSF9, JSF12 and JSF15 as *Podo*- and vibriophage JSF10 as *Siphoviridae*. Our data supports the assignment of phages JSF9, JSF15, and JSF 10. However, JSF12 according to the profile HMM hits should be classified as *Siphoviridae* (Figure 2A,B).

Whole genome alignments revealed that all phages that have been assigned by the ClassiPhage method to an ICTV taxon comprise large genome regions that can be aligned to corresponding classified reference genomes. However, in some cases, the overall coverage of the alignable parts of the phage genomes to reference genomes is sparse. In the case of phage JSF12, experimental data indicates an assignment to *Podoviridae* while the alignment reveals a higher similarity to reference genomes from the *Siphoviridae*. The latter result is in accordance with the results of the profile HMM scan. Both sequences have been aligned and closely inspected using ACT where no missing ORF was observed.

The application of the method on the unclassified vibriophages dataset explored the capabilities of ClassiPhage, where transmission electron micrographs (TEM) images confirm the generated classification. The HMMs of the different families demonstrated a high specificity, meaning that when a phage genome is specifically targeted by HMMs of one family, the HMMs of other families show only insignificant numbers of HMM/protein matches. This specificity further supports the idea that it is possible to use the generated HMMs as a means of classification as discussed by [15].

The generated *Vibrio* derived profiles scanning the proteomes of the phages of the four families gave us the unique opportunity for a Markov based classification, and sometimes subclassification of distantly related phages, independently of shared molecular markers or pairwise alignment, but still in accordance with the ICTV classification scheme.

### 3.5. Inoviridae Taxonomy Phages and Profile HMMs

The nine HMMs specific for *Inoviridae* infecting *Vibrionaceae* were used to scan proteins encoded by all known *Inoviridae*. Profile HMMs scan resulted in a number of positive matches (Appendix A) reflecting that the *Inoviridae* phages infecting *Ralstonia*, *Enterobacteria*, *Pseudomonas*, *Xanthomonas*, and *Stenotrophomonas* encode proteins of the same families as the *Inoviridae* infecting *Vibrionaceae* (Figure 4).

Four out of the nine vibriophage generated HMMs had hits only to *Inoviridae* infecting *Vibrio* hosts proteomes. The rest matched to proteins from non-*VibrioInoviridae*. Although all investigated *Inoviridae* genome encodes more than one *vibrio Inoviridae* like protein, not a single protein family was present in all phages. The most commonly shared protein family members are zot-like proteins, which have been found in 95% of all phages [28]. According to Mai-Prochnow et al. [28] the genomes of *Inoviridae* range within a size of 4 Kbp to 12 Kbp which gives spaces to encode up to 11 genes. The *Inoviridae* profile HMMs generated within this work contain 19 protein families which explains why not each HMM finds a protein in each *Inoviridae* genome supporting the contribution to virulence of the phage class [44]. However, what is indicative for a member of *Inoviridae* is the set of proteins that are found exclusively in members of this phage family [28].

The generated *Inoviridae Vibrio* derived profiles scanning the proteomes of all *Inoviridae* phages gave us the unique opportunity to explore the extent to which proteins are shared between *Inoviridae* infecting different bacterial hosts.

### 3.6. Taxonomy of Podoviridae

To elucidate the taxonomic relation of *Podoviridae* identified by profile HMMs, an extended scan with the *Vibrio Podoviridae* models were performed against a set of *Podoviridae* that infect other bacterial hosts (Figure 5). 

The *Podoviridae* profile HMMs from vibriophages exhibited, as in the case of the *Inoviridae*, hits to multiple proteins out of all published *Podoviridae* phages (Appendix A). *Podoviridae* represent a much more complex and diverse class of phages compared to *Inoviridae*. The genomes of *Podoviridae* from *Vibrionaceae* comprise 96 distinct protein families. However, when grouped by shared HMM hits and hosts the results of the scan display a degree of specificity and sensitivity that may be useful to subclassify the taxon. It is no surprise that phages that prey on the same host share proteins. However, the scores of the HMM hits reflect the degree of similarity shared by the single proteins. Thus, the heatmap shows the diversity of the different protein classes and thus gives us an idea of the phylogenetic history of the proteins.

### 3.7. CDS Prediction and Additional HMM Refinement

The genome annotation of public available phages is the product of gene prediction programs with different sensitivity [45,46,47,48]. This results in genomes where some CDS have not been annotated. To examine the value of HMMs to identify such missing phage CDS, the intergenic regions of each phage genbank file used in this study was scanned using the profile HMMs. In total, 234 nucleotide regions were identified encoding gene products that align to one of the protein families modelled by the HHMs (Appendix A). Indeed, profile HMMs can be used to identify missing CDS. 

To investigate whether these new proteins may improve the profile HMMs, we generated refined HMMs using the original proteins plus the new identified CDS as described in the material and methods section. An evaluation of the refined HMMs identified exactly the same proteins per HMM with slightly moderated hit scores. The test revealed that the refinement of the HMMs did not yield better performing HMMs. The sensitivity of HMMs is correlated much stronger to the diversity than to the number of the proteins used in the initial alignment step. We concluded that our original profile HMMs already contain sufficient diverse proteins to model the protein families and thus the model’s predictive power is already close to saturation.

### 3.8. Identification and Classification of Prophages within Bacterial Genomes

#### Scan of Positive Dataset of Vibrio Genomes

Apart from phage genomes generated from phage particles that have been experimentally confirmed to infect bacteria, host genomes themselves contain in many cases integrated prophages derived from old infection events [49]. To examine the reliability of the profile HMMs with regards to their ability to identify and support the classification of bacteriophages integrated within a bacterial genome, a scan of 10 sequenced *Vibrio* strains with experimentally proven active *Inoviridae* prophages [29] was performed. The bacteriophage family specific HMMs were used to search for matches within the complete protein sets of nine *Vibrio alginolyticus* and one *Vibrio typhli* genome. The same strains have been scanned using PHASTER for phage identification. Whenever HMM hits co-localized and matched a prophage region predicted by PHASTER, they were represented in a separate facet (Figure 6).

In each of the genomes, the profile HMMs hits indicate the presence of genes encoding putative phage proteins. In the case of the strains *V. alginolyticus* K04M1 and K05K4 two complete replicons are present as extra-chromosomal phages [29] where the nine refined HMMs had matches. In all nine *V. alginolyticus* strains, *Inoviridae* derived profile HMMs match to a single locus on chromosome 2 of eight strains, and two other loci on the K09K1 strain.In some instances we could identify two distinct prophages that integrated in close proximity within the host chromosome [29] and was reflected by multiple hits of the same HMM in the same region. While strains K06K5 and K10K4 had an additional *Inoviridae* integrated at the same locus on chromosome 1. For the *V. alginolyticus* strains it has been shown by the Phage-seq method [50] that the corresponding genome regions express biological active *Inoviridae* particles encoding the protein sequences that match the profile HMM.

The scan of a positive data set of 9 *V. alginolyticus* and 1 *V. typhli* genome confirmed several hits for *Inoviridae* proteins, where the integrated prophages were located and experimentally confirmed as well as on three extra-chromosomal *Inoviridae* phages supporting the reliability of the method.

### 3.9. PHASTER and ClassiPhage Scan of Published VibrioGenomes, Commonly and Additional Identified Phage Regions

PHASTER scan of 158 published closed *Vibrio* genomes resulted in the prediction of 458 prophages, out of which 143 were confirmed by the ClassiPhage scan (Appendix A). Additionally, 64 regions where more than three consecutive HMM hits have been predicted by ClassiPhage that indicate protein genes of phage origin (Appendix A). In addition to locus identification, ClassiPhage enabled us to taxonomically classify the prophages into *Ino-*, *Myo-*, *Podo-,* or *Siphoviridae* (Appendix A, Appendix A). Most phages (>90%) could be classified into *Inoviridae* and some as *Podoviridae*. Our results further support the findings that *Inoviridae* are the most frequent phages infecting *Vibrio* species [21]. For *Myoviridae* and *Siphoviridae* HMM hits of one hypothetical protein it is not enough to classify. 

On the other hand, the ClassiPhage method failed in identifying a set of 315 regions predicted by PHASTER. This set of genome regions encode proteins that match to proteins of phages infecting *Salmonella*, *E. coli*, *Bacilli* (Appendix A, Appendix A) such as integrases, recombinase as well as proteins of unknown functions. Note that phages share these kinds of proteins with other types of mobile genetic elements and that PHASTER characterized the vast majority of these loci as incomplete. However, in the case of a vibriophage reference in 61 cases, no HMM generation was possible due to the low number of proteins clustered during the HMM generation steps (vibriophage 12A4, vibriophage 12B12, *Vibrio* 8, *Vibrio* K139, *Vibrio* kappa, Vibrio N4, *Vibrio* pYD38, VfO3K6, Vf33, VfO4K68, VHML, VP4, VP882, VvAW1, X29). 

Future developments, to overcome this limitation, would include using starting data sets not limited to vibriophages, rather using all available phages, generating HMMs and scanning diverse bacterial genomes. The possibility to generate more diverse and inclusive HMMs increases when more clusters generated out of closely related yet diverse phages are used, which reinforces the need to develop a method including more phage sequences, not limited to a host.

Additionally, the HMM scan resulted in hits that could not be assigned to a reference phage family. This might be evidence for vibriophages of so far unknown phage taxa or indicate false positive hits of the ClassiPhage method reflected by a low bit-score value, or due to HGT whenever the bit-score value was high. The scan of published *Vibrio* genomes generates much more hits than to a phage region, the reason why the combination of consecutive hits located in a certain region, size of the identified ORFs, annotation, E-value and bit scores are key to identify which hits belong to a phage and which do not. Hits not belonging to a prophage generally have low scores. In the case of a high score, proteins are annotated as “polymerases” or “flagellum” or “Transposon area”, whereas phage related annotations are explained by being remnants of phages or by HGT.

The use of a combination of profile HMM hits for phage classification is a relatively new approach for the characterization of bacteriophages and thus further steps must be considered to better exploit the method [15].

## 4. Conclusions

In this work, we describe ClassiPhage, a method for phage classification independent of a shared molecular marker, based on combination of multiple profile HMM hits generated from a set of classified phage proteomes, and thus generating a Markov-based classification fitting the ICTV classification. We discussed the generation and refinement of profile HMMs, their validation across four different viral taxa and their application for viral taxonomic classification, focusing on vibriophages. Additionally, we used the generated HMMs to scan whole genomes and benchmarked the identified regions to PHASTER predicted prophage regions, to attempt viral identification prior to classification using the ClassiPhage method. We were able to show that the ClassiPhage method was able to reliably classify, by scanning the protein coding sequences of (i) a set of unclassified vibriophages; (ii) experimentally proven *Inoviridae*; and (iii) integrated phages in a set of closed and published *Vibrio* genomes, into one of the four phage families. We were also able to show that the method is not limited to vibriophages but the potential of the method extends towards phage subclassification, especially in the case of *Podoviridae*. This analysis supports the correlation of the generated HMMs per vibriophage family to the bacterial host. Lastly, we were able to show the potential of the method to be used as a phage identification and classification tool by scanning bacterial genomes using the refined HMMs and analyzing the protein sequence hits with regards to their consecutive location in the host genome. This method showed limitations for the case when scanned unclassified phages had one ambiguous hit to the refined HMMs and when phages identified by PHASTER which were missed by the ClassiPhage method. Phage identification must be coupled with sequence features for correct phage boundary identification. This limitation is a consequence of the quality and the constraints of the HMMs generation step, which makes it clear that fundamental steps must be considered to generate better and more comprehensive viral derived refined HMMs. We foresee that, with an ever-increasing amount of viral sequences and with the generation of robust and comprehensive viral HMMs, this method has the ability to classify phages into their taxonomic family in accordance with the ICTV scheme. The generated scans can subsequently be used in machine learning approaches to automatically classify viral sequences.

## Figures and Tables

**Figure 1 viruses-11-00195-f001:**
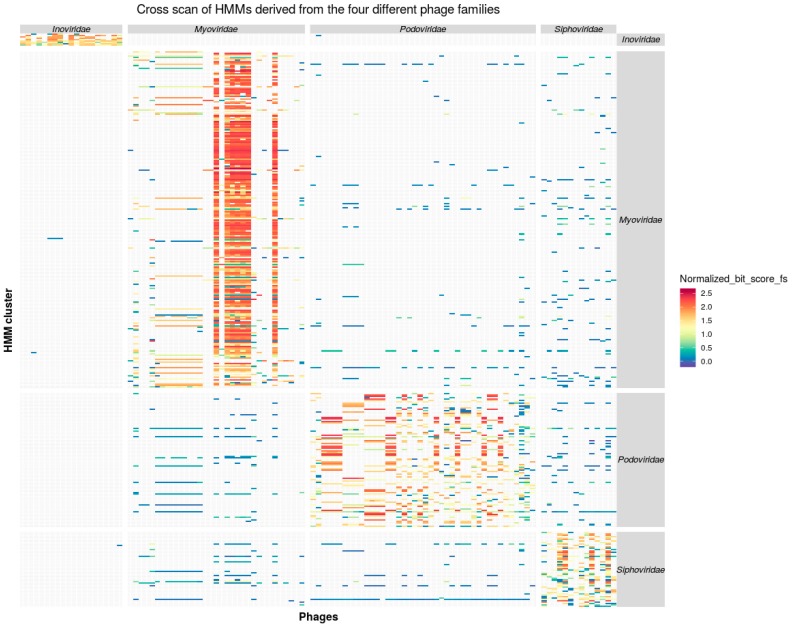
Markov Models (HMM) scan of phage family derived models own input “CDS” and coding sequences of other families. The scan of the protein sequences derived from *Ino*-, *Myo*-, *Podo*-, and *Siphoviridae*, was conducted by the profile HMMs. The names of all phages grouped into phage-families are marked at the bottom of heatmap. The bit-score of the HMM matches was normalized by the size (in bp) of the HMM’s consensus sequence (data see Appendix A). The results are color-coded from blue (low-score) to red (high-score).

**Figure 2 viruses-11-00195-f002:**
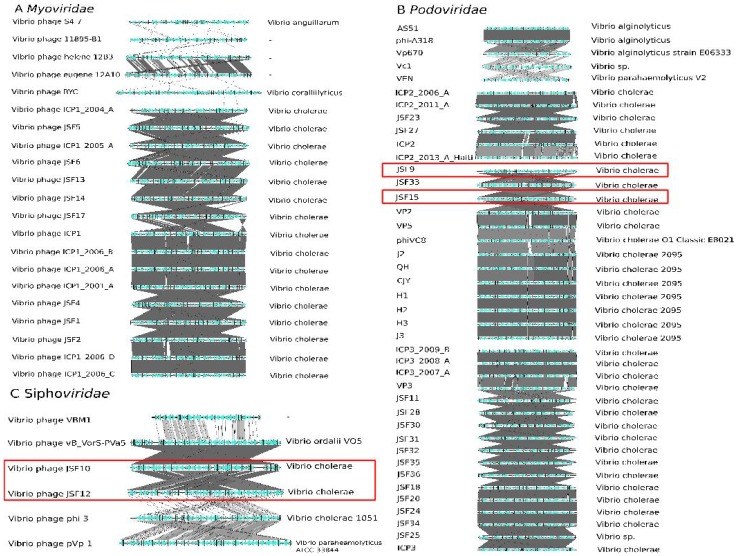
Alignment of *Caudovirales* genomes. (**A**) *Myoviridae*, (**B**) *Podoviridae*, and (**C**) *Siphoviridae*. Genomes of phages that have not yet been assigned by ICTV are marked in pink. Four phages JSF9, JSF10, JSF12, and JSF15 are boxed in red. JSF12 has been assigned to *Podoviridae* based on transmission electron micrographs (TEM) the complete genome alignment indicates a close relation to the *Siphoviridae* phage JSF10. The data has been visualized with Easyfig.

**Figure 3 viruses-11-00195-f003:**
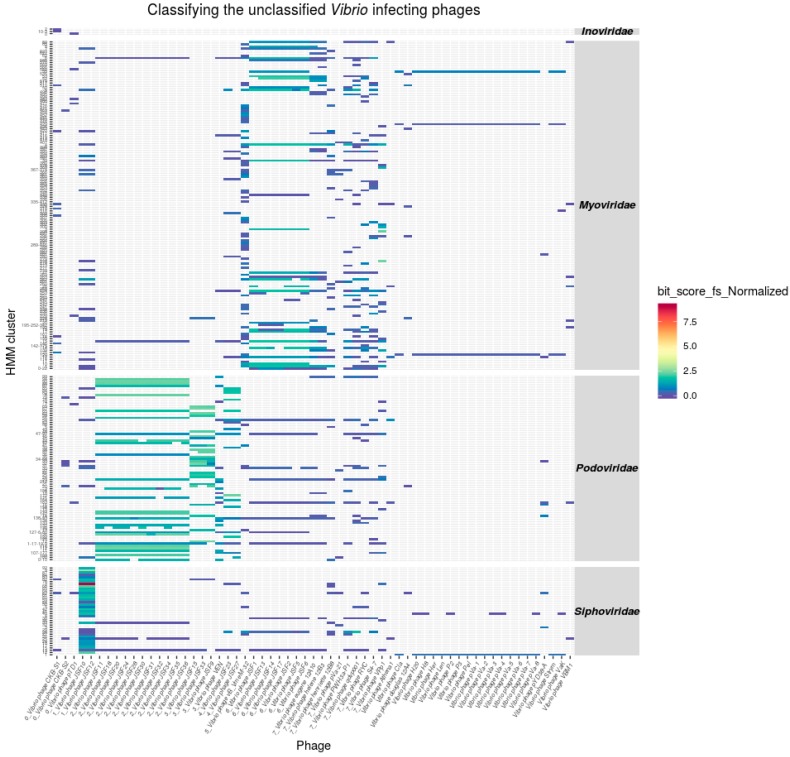
Taxonomic classification of vibriophages. This heatmap shows a profile HMM scan on the proteins of 58 unclassified bacteriophages genomes. Forty-one unclassified genomes generated sufficient with enough hits to be assigned to a taxonomic group. The HMMs have been integrated in the heatmap (*x*-axis). The HMMs are grouped (on the *y*-axis) into the respective phage families. The indicator for the quality of a hit is color coded to the normalized bit-score assigned for the respective match by hmmscan.

**Figure 4 viruses-11-00195-f004:**
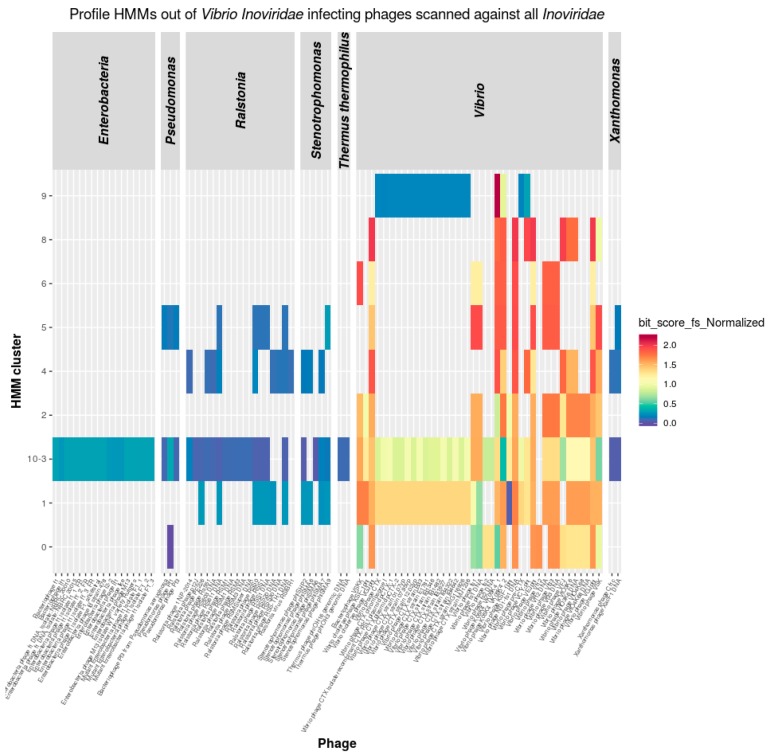
HMM scan results of all *Inoviridae* phages.This heatmap shows an *Inoviridae* derived profile HMM (*y*-axis) scan on the proteins of 119 *Inoviridae* genomes grouped by host genome (*x*-axis). HMMs ranged from hits specific to *Inoviridae* infecting *Vibrionaecea* to general hits for *Inoviridae* infecting other hosts. The indicator for the quality of a hit is color coded to the normalized bit-score assigned for the respective match by hmmscan.

**Figure 5 viruses-11-00195-f005:**
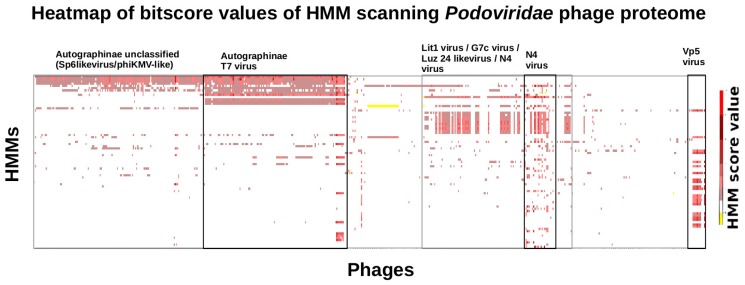
Profile HMM scan of *Podoviridae* HMMs from *Vibrionaceae* versus genomes from *Podoviridae* phages infecting non-vibrio hosts. This heatmap shows a profile HMM scan on the proteome of 1066 *Podoviridae* genomes. Sufficient hits were generated to discriminate four groupings of *Podoviridae.* The HMMs have been integrated in the heatmap (*y*-axis). The HMMs are grouped (on the *x*-axis) into general *Podoviridae* subclassifications. The indicator for the quality of a hit is color coded to thenormalized bit-score assigned for the respective match by hmmscan. The generated hmmscan output was visualized using matplotlib library in Python 3.5.

**Figure 6 viruses-11-00195-f006:**
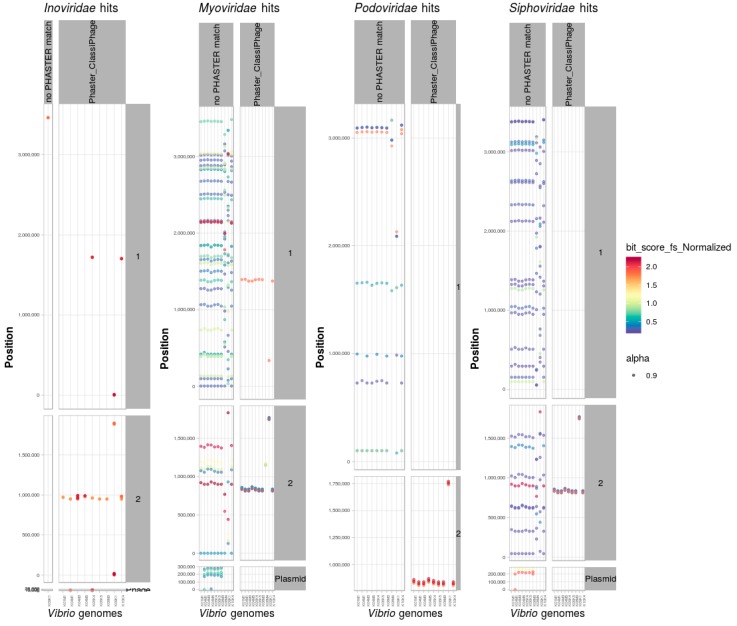
HMM search for prophages in *Vibrio* genomes with proven phage activities.Family specific HMMs constructed for *Ino*-, *Myo*-, *Podo*-, *andSiphoviridae* (grouped on *x*-axis) were used to scan all proteins derived from the genome of nine *V. alginolyticus* and one *V. typhli* genomes (*x*-axis per phage family grouping). In all of the *V. alginolyticus* genomes, regions encoding proteins matching to the profile HMMs were found (plotted per position and grouped per replicon on the *y*-axis). In cases where a region with consecutive HMM hits predicted as well by PHASTER was separately faceted.

**Table 1 viruses-11-00195-t001:** Phage family specific HMMs *.

	No of Genomes (Size in Kbp)	No of Proteins	Proteins after MCL	HMMS with >5 Proteins	Positive Evaluated HMMs
Siphoviridae	14 (37.3–128.6)	1497	414	94	54
Podoviridae	42 (38.4–112.1)	2641	490	233	96
Myoviridae	35 (33.1–250)	5915	921	634	242
Inoviridae	19 (6.3–21)	241	39	12	9
Total	110	10,294	1864	973	401

* Details on the complete calculation of the models are in Appendix A.

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
