# Peer review of "Classifying the Unclassified: A Phage Classification Method"

_viruses, 2019, doi:10.3390/v11020195_

Round 1
Reviewer 1 Report
In the study by Chibani and colleagues, profile HMMs are used to assess the classification of bacteriophages infecting bacteria of the family Vibrionaceae, and this methodology is used to assess the classification of the phage family Podoviridaeand to search for prophages in a set of Vibrio genomes. The authors describe this as the method IdentiPhage and suggest it can be used in official classification by ICTV.
I have several major issues with the justification and methodology of this study.
1. It is clear that the authors are knowledgeable about bioinformatics, but much less so about bacteriophage classification. From the introduction, it sounds like this study is meant to aid the ICTV in the classification of new bacteriophages. However, they have missed out on the majority of phage-specific literature that has been published on this topic in the last couple of years. This is important, as this is not the first study that uses a form of protein clustering to analyse phage classification. I suggest to have a look at the following references:
a. Phage Proteomic Tree: Rohwer & Edwards, 2002
b. Prokaryotic Virus Orthologous Groups: Grazziotin et al, 2017
c. vConTACT: Bolduc et al, 2017
d. GRAViTy: Aiewsakun et al, 2018
e. Latest Bacterial and Archaeal Virus Subcommittee updates in Archives of Virology for an idea of where phage classification is headed. (please know that the Subcommittee members are always happy to consult on any phage classification-related query)
f. Phage classification guide: Adriaenssens & Brister, 2017
2. What made me initially excited about this manuscript, is the availability of a novel method or tool. However, the authors have not made their methods or HMM profiles publicly available. As such, the title and introduction of the paper create a false expectation with the reader that “IdentiPhage” is something they can download or use for themselves. In reality, this study presents a profile HMM-based assessment of Vibrio phage diversity with potential applications for future phage classification.
3. My third comment is methodological. The authors state in the Materials & Methods section that they worked with publicly available sequences and used the GenBank files to extract the CDSs. This method can introduce a bias when different qualities of annotations are combined. For example, if one of the genomes does not have a certain CDS predicted (while it is there), this will create a false negative result in the HHM profile.
4. If I understand it correctly, the HMM profiles were only generated from the Vibrio-infecting phage dataset. Based on those data, 44 out of 58 unclassified genomes were put into one of the three tailed phage families. Why was this not done with profiles from the entire order? Isn’t it possible that these genomes could be easily classified based on a closest relative that does not infect Vibrios? For example, KVP40 was first classified based on its relation to E. coli phage T4, and later on put into its own taxon.
5. The analysis of the Podoviridaephages with the Vibrio phage-specific HMMs mostly disregards the classification that is already in place, as it only mentions the subfamily Autographivirinae and does not include the subfamilies Picovirinaeand Sepvirinae, and about 20 genera. In addition, as in point 4, I do not think a Vibrio phage HMM profile is appropriate to analyse the whole family.
6. The authors have used their HMMs to investigate potential prophages in bacterial genomes. This is an area of research where new methods and tools are needed. However, when compared to the tool PHASTER, IdentiPhage only really seems to be useful in finding inoviruses. Why should one go through the effort of using an elaborate method as described here, when PHASTER vastly outperforms it?
7. My last major point is one of style and punctuation. There are quite a few grammar mistakes left in the manuscript, and in many cases punctuation is lacking. Please use more commas.
Minor & specific comments
The plural of taxon is taxa, not taxons.
Lines 173-174: This statement is not exactly warranted using these data. The authors used members of the Caudovirales(dsDNA phages) and Inoviridae(ssDNA phages). Naturally, the Caudoviralestaxon will show as being monophyletic.
In addition, the current families within the order Caudovirales are most definitely not monophyletic, creating some confusion when reading this statement.
Figure 2: I’m not quite sure how this figure contributes to the story of the paper.
Line 313: put reference in correct citation format
Line 349 and on: If more Vibrio prophages show similarity to non-Vibrio phages than to other Vibrio phages, how can this support the correlation between phage and host? I do not understand the explanation provided here.
Author Response
We thank the reviewer for the precise and detailed work on our manuscript. We are sure that it helped us to improve the manuscript substantially. We hope that we could address each issue apropriately.
Major issues:
(1)
It is clear that the authors are knowledgeable about bioinformatics, but much less so about bacteriophage classification. From the introduction, it sounds like this study is meant to aid the ICTV in the classification of new bacteriophages. However, they have missed out on the majority of phage-specific literature that has been published on this topic in the last couple of years. This is important, as this is not the first study that uses a form of protein clustering to analyse phage classification. I suggest to have a look at the following references:
a. Phage Proteomic Tree: Rohwer & Edwards, 2002
b. Prokaryotic Virus Orthologous Groups: Grazziotin et al, 2017
c. vConTACT: Bolduc et al, 2017
d. GRAViTy: Aiewsakun et al, 2018
e. Latest Bacterial and Archaeal Virus Subcommittee updates in Archives of Virology for an idea of where phage classification is headed. (please know that the Subcommittee members are always happy to consult on any phage classification-related query)
f. Phage classification guide: Adriaenssens & Brister, 2017
Reply:
First of all we agree that the introduction might be misleading. The aim of this study was to evaluate the power of combined Markov models to generate a taxonomic assignment of phages based on genome sequences. We emphasized on the question whether this kind of taxonomic assignments is in agreement with ICTV. Of course there are other studies that used clustering of proteins to generate a phage taxonomy. Though we are aware of non-HMM clustering methods we focused in our introduction on the HMM based methods. We are grateful for the suggested references and reworked the introduction accordingly.
(2)
What made me initially excited about this manuscript, is the availability of a novel method or tool. However, the authors have not made their methods or HMM profiles publicly available. As such, the title and introduction of the paper create a false expectation with the reader that “IdentiPhage” is something they can download or use for themselves. In reality, this study presents a profile HMM-based assessment of Vibrio phage diversity with potential applications for future phage classification.
Reply:
We agree completely with the reviewer. We investigated whether profile-HHMs allow an assessment of vibrio phages. We provide the HMMs available for download (https://owncloud.gwdg.de/index.php/s/3oYAa4vb0yz17M8). It is true that the name “IdentiPhage” raises the expectation of an identification tool. Since we want to show that the combination of HMMs is powerful enough to classify an already known phage genome we rename the method to “ClassiPhage”.
Indeed the results of this study suggest that it might be possible to develop a tool that meets the expectations of the reviewer. This is currently the scope of an ongoing project within our group.
(3)
My third comment is methodological. The authors state in the Materials & Methods section that they worked with publicly available sequences and used the GenBank files to extract the CDSs. This method can introduce a bias when different qualities of annotations are combined. For example, if one of the genomes does not have a certain CDS predicted (while it is there), this will create a false negative result in the HHM profile.
Reply:
We agree with the reviewer that there is a clear diversity concerning the quality of the genome annotation of public available data sets. This is one of the reasons why we limited the investigation to vibrio infecting phages. Since Inoviridae infecting vibrios are investigated since decades there is solid knowledge available on which genes are expected to be present. However, we address the concern of the reviewer and checked the whole set of phages infecting vibrios for missing ORFs and investigated if the profile HMMs can be improved by these proteins. The result of this check is described in the following paragraph. If it is requested we can add this to manuscript.
CDS prediction and additional HMM refinement
The genome annotation of public available phages are the product of gene prediction programs with different sensitivity [40–43]. This results in genomes where some coding sequences (CDS) have not been annotated. To examine the value of HMMs to identify such missing phage CDS, the intergenic regions of each phage genbank files used in this study was scanned using the profile HMMs. In total 234 nucleotide regions have been identified that encode gene products that align to one of the protein families modelled by the HHMs (Table Ign_hmm_scan.csv). In deed profile HMMs can be used to identify missing CDS.
To investigate whether these new proteins may improve the profile HMMs we generated refined HMMs the original proteins plus the new identified CDS as described in the material and methods section. An evaluation of the refined HMMs identified exactly the same proteins per HMM with slightly moderated hit scores. The test revealed that the refinement of the HMMs did not yield better performing HMMs. The sensitivity of HMMs is correlated much stronger to the diversity than to the number of the proteins used in the initial alignment step. We concluded that our original profile HMMs already contain sufficient diverse protein to model the protein families and thus the models predictive power is already close to saturation.[1]
(4)
If I understand it correctly, the HMM profiles were only generated from the Vibrio-infecting phage dataset. Based on those data, 44 out of 58 unclassified genomes were put into one of the three tailed phage families. Why was this not done with profiles from the entire order? Isn’t it possible that these genomes could be easily classified based on a closest relative that does not infect Vibrios? For example, KVP40 was first classified based on its relation to E. coli phage T4, and later on put into its own taxon.
Reply:
The reviewer is right, we selected exclusively Vibrio infecting phages as a test data set. We did not include the whole order of the available phages for three reasons: (i) the number of available phages infecting vibrios already generated enough diversity to produce valid and sensitive HMMs. (ii) as the reviewer pointed out in concern (3): since the quality of genome annotation is quite diverse in the public available data the risk of using wrong annotated input genomes increases with the number of used data sets. Especially if one includes completely new phages where supervised gene prediction programs tend to perform with an increased error rate. (iii) using reference members of the phage taxon requires prior knowledge about the appropriate best matching member with the highest genome annotation quality of the taxon. If this kind of knowledge is available there might be cases that such a supervised prediction approach may outperform our approach. However, the de novo clustering of ClassiPhage, that can be used with any given set of phage genomes with a sufficient internal diversity, does not require any prior knowledge and thus is an unsupervised prediction method.
(5)
The analysis of the Podoviridaephages with the Vibrio phage-specific HMMs mostly disregards the classification that is already in place, as it only mentions the subfamily Autographivirinae and does not include the subfamilies Picovirinaeand Sepvirinae, and about 20 genera. In addition, as in point 4, I do not think a Vibrio phage HMM profile is appropriate to analyse the whole family.
Reply:
This is true, we did not refer to the already existing classification at all. The reviewer is completely right, Vibrio phage HMMs as exclusive input source are not appropriate to classify the whole Podoviridae. Caudovirales including Podoviridae do not represent a monophyletic group, our aim was to use as a starting point for the evaluation a most likely defined subgroup of closely related virus genomes and to investigate the power of profile HMMs to classify member of such a group and in a second step to extend the set of genomes to see how more distant related or even not related genomes can be classified by these initial models. Even though we used this limited input, that is heavily biased to vibrio infecting phages, it was still possible to recover some of the subfamilies. Essentially those subfamilies that contain enough members with sufficient internal diversity within the input data, like the Autographivirinae.
This strongly supports the discriminative power of profile –HMMs. Our studies demonstrates that a valid classification of the Podoviridae could start with a group of heavily biased profile-HMMs. In our case those preying on Vibrios, and may be extended using findings achieved with theses models outside the input data set. However, we did not intend to do a taxonomic investigation of Podoviridae in total but are happy to describe an experiment in an outlook part of the paper how such an investigation could be done, if this is requested.
(6)
The authors have used their HMMs to investigate potential prophages in bacterial genomes. This is an area of research where new methods and tools are needed. However, when compared to the tool PHASTER, IdentiPhage only really seems to be useful in finding inoviruses. Why should one go through the effort of using an elaborate method as described here, when PHASTER vastly outperforms it?
Reply:
The scope of our investigation was to investigate how correct profile-HMMs classify known phages respectively prophages. However, a profile-HMMs scan of a genome can be used as well to identify a prophages locus in case of clustered hits. In contrast PHASTER is designed to identify prophages in genomes and generates as a byproduct a suggestion of a taxonomic origin. A comparison of the results show that ClassiPhage outperforms PHASTER concerning the taxonomic classification whereas PHASTER generates more phage loci prediction. However, a closer look at the high number of PHASTER prediction reveals that they contain a high number of incomplete phages and remnants of phages such as recombinases, integrases and so on.
So to answer the reviewers question: both program perform superior in the scope of their design. So if one wants taxonomic classification use ClassiPhage if you want to know as much phage loci as possible use PHASTER.
(7)
We corrected the English wording with the help of a native speaker.
Minor issues:
The plural of taxon is taxa, not taxons.
True, we corrected that.
Lines 173-174: This statement is not exactly warranted using these data. The authors used members of the Caudovirales(dsDNA phages) and Inoviridae(ssDNA phages). Naturally, the Caudoviralestaxon will show as being monophyletic.
In addition, the current families within the order Caudovirales are most definitely not monophyletic, creating some confusion when reading this statement.
We admit that our statement is misleading or even wrong. It is for sure not possible to say anything about the mono- or the polyphyletic nature of the complete phage families using the investigated data. Actually, we are aware of the fact that Caudovirales are a polyphyletic taxon. However, if one reduces Caudovirales to the subset of Caudovirales infecting Vibrio and compares those genomes on the complete nucleotide level the found similarities indicate the presence of monophyletic groups, as is visible in figure 2. We rewrote the incriminated passage accordingly
Figure 2: I’m not quite sure how this figure contributes to the story of the paper.
See the remarks above.
line 313: put reference in correct citation format
done
Line 349 and on: If more Vibrio prophages show similarity to non-Vibrio phages than to other Vibrio phages, how can this support the correlation between phage and host? I do not understand the explanation provided here.
PHASTER predicts the phage nature of a given locus if it contains key genes typical for phages and gives in addition a vote whether the locus encodes a complete or an incomplete locus. It is known from the genome analysis of mobile genetic islands that prophages, IS-elements and transposons share genes that may mislead PHASTER to annotate a smaller type of a mobile genetic element as an incomplete prophage. We think that this is in most of the additional findings of PHASTER true since these elements have in the vast majority been characterized as incomplete. We rewrote the passage accordingly.
Reviewer 2 Report
Phages are the most abundant biological entities in the earth. The international Committee on Taxonomy of Viruses (ICTV) governed the taxonomic classification of viruses based upon phage properties including virus genome, the structure of viral capsid, host range, pathogenicity, and sequence similarity. Classification of viruses based on NGS sequence has been challenged due to the lack of a universal marker gene and the high polyphyleticity of phages.
In this manuscript, they classified the vibrio phages using the IdentiPhage, a method for phage classification based on combination of multiple profile Hidden Markov Model (HMM) generated from as set of classified phage proteomes.
The IdentiPhage method generated 401 HMM profiles from each different vibriophage (Sipho-, Podo-, Myo- and Inovirdae) genomes.
The generated profile HMMs was applied to unclassified 58 vibrio phage genomes. The scan resulted in the assignment of 44 out of 58 genomes to vibriophages families (Sipho-, Podo-, Myo- and Inovirdae).
They also applied the generated profile HMMs to Vibrio genomes to identify vibriophages and found consistent results using PHASTER for phage identification.
They demonstrated that the IdentiPhage method can reliably classify; i) unclassified vibriophages; ii) experimentally proven Inoviridae, the most common vibriophage; iii) vibriophages genomes integrated in host chromosomal DNA.
This manuscript is well-written and easy to follow. Despite of overwhelming information from NGS data, we were not able to classify or define phages integrated in the host chromosome. The IdentiPhage would definitely contribute to characterize and classify existing and new phages from Vibrio or other bacterial species. With a growing interest in utilization of phage to biomedical application, this is an important manuscript.
Author Response
Dear Teviewer,
thank you for working on our manuscript. We are happy that you like the article. The complete manuscript has been spell checked by a native speaker. In deed we could remove a number of typos and add some missing commas.
thank you again for your time
Round 2
Reviewer 1 Report
With the new angle and the clarifications to this paper, the authors have properly improved the manuscript.
Thank you for making the HMMs available to the community. Please do make them public upon acceptance, as the link leads to the general software page of the group.